# Human induced pluripotent stem cell-derived three-dimensional cardiomyocyte tissues ameliorate the rat ischemic myocardium by remodeling the extracellular matrix and cardiac protein phenotype

Junya Yokoyama[1], Shigeru Miyagawa[1], Takami Akagi[2], Mitsuru Akashi[2], Yoshiki Sawa[1]*

1 Department of Cardiovascular Surgery, Osaka University, Graduate School of Medicine, Osaka, Japan,
2 Building Block Science Joint Research, Graduate School of Frontier Biosciences, Osaka University, Osaka, Japan

* sawa-p@surg1.med.osaka-u.ac.jp

## Abstract

The extracellular matrix (ECM) plays a key role in the viability and survival of implanted human induced pluripotent stem cell-derived cardiomyocytes (hiPSC-CMs). We hypothesized that coating of three-dimensional (3D) cardiac tissue-derived hiPSC-CMs with the ECM protein fibronectin (FN) would improve the survival of transplanted cells in the heart and improve heart function in a rat model of ischemic heart failure. To test this hypothesis, we first explored the tolerance of FN-coated hiPSC-CMs to hypoxia in an *in vitro* study. For *in vivo* assessments, we constructed 3D-hiPSC cardiac tissues (3D-hiPSC-CTs) using a layer-by-layer technique, and then the cells were implanted in the hearts of a myocardial infarction rat model (3D-hiPSC-CTs, n = 10; sham surgery control group (without implant), n = 10). Heart function and histology were analyzed 4 weeks after transplantation. In the *in vitro* assessment, cell viability and lactate dehydrogenase assays showed that FN-coated hiPSC-CMs had improved tolerance to hypoxia compared with the control cells. *In vivo*, the left ventricular ejection fraction of hearts implanted with 3D-hiPSC-CT was significantly better than that of the sham control hearts. Histological analysis showed clear expression of collagen type IV and plasma membrane markers such as desmin and dystrophin *in vivo* after implantation of 3D-hiPSC-CT, which were not detected in 3D-hiPSC-CMs *in vitro*. Overall, these results indicated that FN-coated 3D-hiPSC-CT could improve distressed heart function in a rat myocardial infarction model with a well-expressed cytoskeletal or basement membrane matrix. Therefore, FN-coated 3D-hiPSC-CT may serve as a promising replacement for heart transplantation and left ventricular assist devices and has the potential to improve survivability and therapeutic efficacy in cases of ischemic heart disease.

**Data Availability Statement:** All relevant data are within the manuscript and its Supporting Information files.

**Funding:** This research was partially funded by JSPS KAKENHI Grant Number JP18K16391. https://kaken.nii.ac.jp/grant/KAKENHI-PROJECT-18K16391/ No additional external funding was received for this study.

**Competing interests:** The authors have declared that no competing interests exist.

## Introduction

Despite recent developments in novel drug therapies for heart failure [1], surgical treatments such as implantation of a left ventricular assist device and heart transplantation remain the last lines of defense for heart failure patients. However, major complications of these procedures, including a high risk of device-induced infection or cerebral hemorrhage [2], and donor shortages [3] limit the application of these interventions. For these reasons, regenerative medicine such as cell therapy has been aggressively introduced to treat heart failure.

With these efforts, new cardiac constructs that include three-dimensional (3D) human induced pluripotent stem cell-derived cardiomyocyte (hiPSC-CM) tissue, repeated transplantation of layered cell sheets [4], the mixing of vascular endothelium to construct a tissue [5], and creating tissue with blood vessels *in vitro* [6] have been developed. However, these methods have proven to be complicated and time-consuming. In our previous study, we developed a novel coating method with extracellular matrix (ECM) nanofilms [7, 8]. However, the therapeutic efficacy and the specific cardiac proteins or ECM remodeling effects that contribute to cardiomyogenesis in heart tissue regeneration after *in vivo* transplantation remain unclear.

In this study, we prepared myocardial tissues from 3D-hiPSC-CMs using a fibronectin (FN) and gelatin (G) coating as 3D-hiPSC-CM tissues (3D-hiPSC-CTs). FN was selected for this design given its potential for improving the production of various cardiomyocyte proteins via the α6β1 receptor, along with the ischemic resistance of cardiomyocytes [9], and the supply of a scaffold for cell adhesion, which could thereby improve cell survival after transplantation. FN has strong adhesion to vascular endothelial cells and is known to act as an ECM based on endothelial cell recruitment and vascular elongation during angiogenesis, together with cytokines produced by the cells [10]. Thus, fibronectin not only exerts an effect on tissue formation by cell adhesion but is also an essential protein in the early stage of angiogenesis, which could therefore promote cardiac tissue formation. Accordingly, we hypothesized that FN-G-coated 3D-hiPSC-CTs could survive and improve the reduced function of ischemic heart failure in a rat model owing to remodeling of cardiac proteins and the basement membrane matrix. This study aims to identify ECM coated cardiovascular cell populations derived from hiPSC-CM and to generate 3D-hiPSC-CTs with excellent survival, perfusion, and ECM models that can be applied to functional recoveries that are relevant to clinical therapies.

## Materials and methods

### Culture and differentiation of hiPSC-CMs

The hiPSCs (253G1; Riken, Ibaraki, Japan) were cultured and maintained in primate embryonic stem cell medium (ReproCELL, Kanagawa, Japan) with 4 ng/mL human basic fibroblast growth factor (bFGF; Wako, Osaka, Japan) on mouse embryonic fibroblast cells (ReproCELL).

Cardiac differentiation was induced based on a previously reported method developed at our institution [11, 12]. In brief, hiPSC were dissociated using a dissociation solution (ReproCELL) and then transferred to an ultralow-attachment culture dish (Corning, MA, USA) in mTeSR1 (Stemcell Technologies, Canada) with Y-27632 (Wako). After formation of embryoid bodies, the culture medium was replaced with a differentiation medium that contained StemPro34 (Thermo Fisher Scientific, Waltham, MA, USA), 2 mM L-glutamine (Thermo Fisher Scientific), 50 mg/mL ascorbic acid (Wako), and L-thioglycerol (Sigma-Aldrich, St. Louis, MO, USA), which was supplemented with several human recombinant proteins, including bone morphologic protein 4, activin A, bFGF, and vascular endothelial growth factor (VEGF; R&D Systems, MN, USA), and small molecules such as IWR-1 (Wako). The hiPSC-CMs were

maintained in Dulbecco's modified Eagle medium (DMEM; Nacalai Tesque, Kyoto, Japan) containing 10% fetal bovine serum (FBS; Sigma-Aldrich).

## Three-dimensional cardiac tissue formation

Three-dimensional cardiac tissue was constructed using a filtration layer-by-layer technique, which enables the coating of individual cells with ECM [13]. Isolated hiPS-CMs were added to a 6-well culture insert (3.0 μm pore size). The cells were immersed in 2.5 mL of 0.2 mg/mL FN (Sigma-Aldrich) and washed with phosphate-buffered saline (Nacalai Tesque) using a shaking incubator. These cells were then immersed in 2.5 mL of 0.2 mg/mL gelatin (G; Wako) and washed again with phosphate-buffered saline. After nine steps of coating, approximately 10-nm nanofilms of FN-G were coated on each of the cell surfaces (Fig 1A). These cells were suspended in DMEM with 10% FBS and counted using a Countess Automated Cell Counter (Invitrogen, USA). Coated hiPSC-CMs ($3 \times 10^6$ cells) were seeded into 24-well cell culture inserts with a semi-permeable membrane, which was set in a 6-well cell culture plate, followed by the addition of 6 mL of DMEM with FBS and incubation in a 5% $CO_2$ atmosphere at 37˚C. The medium was changed every 24 h. After 4 days of incubation, the samples were evaluated

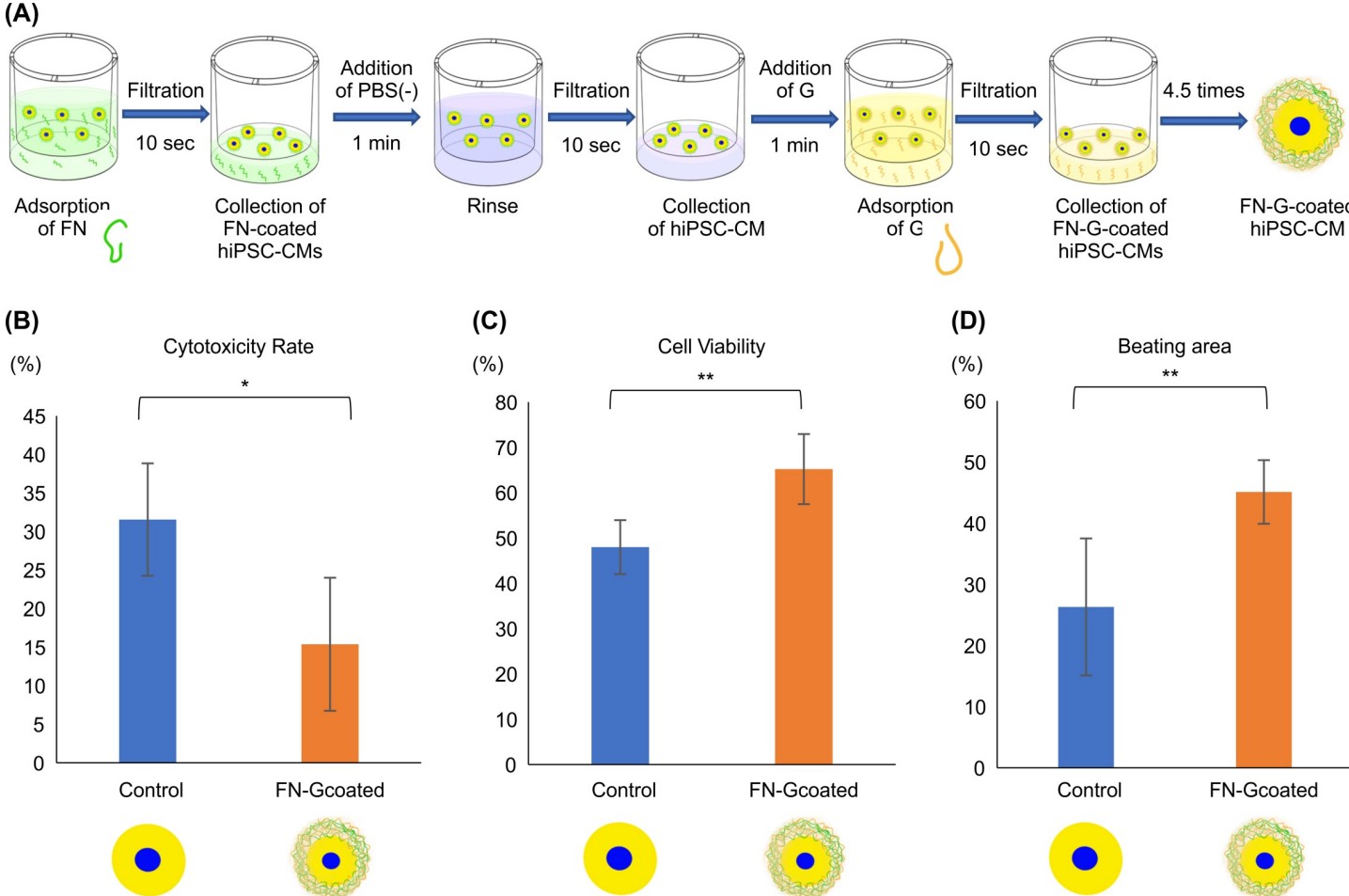

**Fig 1. Fibronectin (FN)-gelatin (G)-coated human induced pluripotent stem cell-derived cardiomyocytes (hiPSC-CMs) had improved tolerance to hypoxia.** (A) Schematic illustration of layer-by-layer filtration for nanofilm coating with FN and G on cell surfaces. (B) Lactate dehydrogenase production (LDH) assay. (C) Cell survival using Cell Counting Kit-8. (D) The beating area of cells assessed by Cell Motion Imaging System. * P < 0.01, ** P < 0.001.

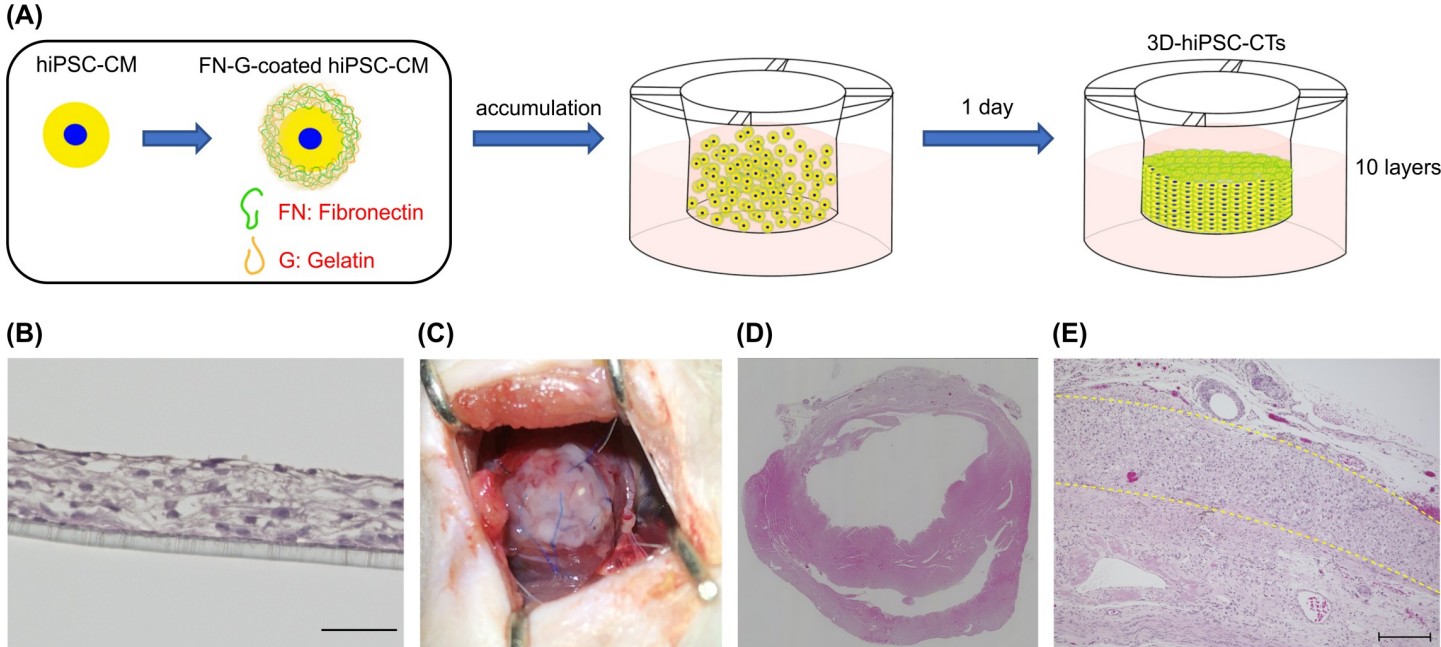

**Fig 2. Transplantation of three-dimensional human induced pluripotent stem cell-derived cardiac tissue (3D-hiPSC-CT) to the infarcted rat heart.** (A) Schematic illustration of construction of 3D-hiPSC-CT by the cell accumulation technique. (B) Hematoxylin and eosin (HE) staining image of 3D-hiPSC-CTs; scale bar = 100 μm. (C) Implantation of 3D-hiPSC-CT to the heart surface at the infarction site. (D) Left ventricular histology assessed 4 weeks after transplantation with HE staining. (E) Engraftment of 3D-hiPSC-CT to the heart surface at the infarction site; scale bar = 100 μm. Dashed yellow lines represent the areas of transplanted 3D-hiPSC-CT.

*in vitro* or implanted in rats. The sample that corresponded to about 10 layers was used as the 3D-hiPSC-CTs (Fig 2A and 2B). We confirmed that the cardiomyocytes were beating in this experimental system daily and that the myocardial tissue itself was beating at the time of transplantation (S1 Movie). FBS was flushed out avoid immunological rejection prior to CT transplantation.

## Cell viability assay

Cell viability was assessed using Cell Counting Kit-8 (CCK8; Dojindo). FN-G-coated and uncoated (control) hiPSC-CMs ($2.5 \times 10^4$ per well) were respectively seeded onto 96-well plates. hiPSC-CMs were cultured at 37°C for 1 day. During the second day, the media were replaced with fresh media and then the cells were incubated under hypoxia (5% $O_2$, 37°C) for 3 days. Subsequently, 10 mL of CCK8 solution was added to each well and further incubated for 2 h at 37°C. Absorbance was recorded on a microplate reader at 450 nm with a reference wavelength of 630 nm. Cell viability was determined as the percentage of surviving cells compared to that of the control. The experiments were performed in triplicates.

## Lactate dehydrogenase (LDH) release assay

The LDH release assay was performed using LDH Cytotoxicity Detection Kit (Takara Bio, Shiga, Japan). FN-G-coated and uncoated hiPSC-CMs ($2.5 \times 10^4$ cells/well) were respectively seeded into 96-well plates and cultured at 37°C for 1 day. The next day, the media were changed, and the cells were incubated under hypoxia (5% $O_2$, 37°C) for 3 days.

A mixture of diaphorase and $NAD^+$ was added to each well. After incubation in a dark room for 30 min at room temperature (20–25°C), the absorbance was recorded on a microplate reader (DS Pharma Biomedical, Osaka, Japan) at 490 nm with a reference wavelength of

600 nm. The experiments were performed in triplicates. LDH release was determined as the percentage of LDH release compared with that of the control.

## Cell motion analyses

The contractile properties of the cells were assessed using a Cell Motion Imaging System (SI8000; SONY, Tokyo, Japan). Videos of the coated hiPSC-CMs were recorded at a frame rate of 150 frames per second, a resolution of $1024 \times 1024$ pixels, and a depth of 8 bits. The hiPSC-CMs were cultured at 37˚C for 1 day. The next day, the media were changed, and the cells were incubated under hypoxia (5% $O_2$, 37˚C) for 3 days.

## Animal care

The animal care procedures were conducted in compliance with the Guide for the Care and Use of Laboratory Animals (National Institutes of Health publication No. 85–23, revised 1996). All animal experimental protocols were approved by the Ethics Review Committee for Animal Experimentation of Osaka University Graduate School of Medicine.

## Myocardial infarction model establishment and 3D-hiPSC-CT implantation

Female F344/NJcl-rnu/rnu rats at 7 weeks of age (Clea Japan, Tokyo, Japan) were anesthetized by inhalation of isoflurane (1.5%; Mylan Inc., Tokyo, Japan), intubated, and mechanically ventilated. The proximal left anterior descending artery at 2 mm below the left appendage was permanently ligated with a 6–0 polypropylene suture (Ethicon, Johnson & Johnson, USA) under left thoracotomy. Two weeks after infarction, transthoracic echocardiography (ViVid i; GE Healthcare, WI, USA) was performed with an 11.5-MHz transducer, and successfully established heart failure model rats [left ventricle ejection fraction (LVEF) < 50%] were selected. The rats were randomly divided into two groups: the 3D-hiPSC-CT group (n = 10) and the sham surgery control group (n = 10). In the 3D-hiPSC-CT group, the CT was implanted on the surface of the infarct zone of the left ventricle, sutured with 7–0 polypropylene, and covered with pericardium and fibrin glue (Beriplast P: CSL Behring, USA) (Fig 2C). In the sham surgery control group, only fibrin glue was added on the surface of the infarct zone of the heart. At 4 weeks after implantation, the rats were sacrificed under general anesthesia with 5% isoflurane inhalation and the heart was promptly dissected. Another five rats that were implanted with the 3D-hiPSC-CT were observed for 12 weeks and then sacrificed appropriately in the same manner.

## Evaluation of cardiac function

Cardiac function was assessed using cardiac echography under general anesthesia (ViVid i; GE Healthcare, WI, USA) with an 11.5-MHz transducer every week until 4 weeks after implantation. The left ventricular end-diastolic (Dd) and end-systolic diameters (Ds) were measured, and the left ventricle end-diastolic volume (LVEDV) and left ventricle end-systolic volume (LVESV) were calculated using the Teichholz formula:

$$LVEDV(ml) = \frac{7}{2.4 + Dd} * Dd^3$$

$$LVESV(ml) = \frac{7}{2.4 + Ds} * Ds^3$$

Left Ventricular ejection fraction (LVEF) was calculated using the following formula:

$$\text{LVEF } (\%) = 100 \times (\text{LVEDV} - \text{LVESV})/\text{LVEDV}$$

## Immunohistochemistry and histology

The sacrificed heart specimens were embedded in paraffin. The paraffin-embedded sections were then stained with hematoxylin–eosin or Picro-Sirius Red to assess the extent of fibrosis. Ten different fields were randomly selected. Percent fibrosis in the infarcted remote zone was calculated as the percentage of pink-colored collagen in the remote area. The paraffin-embedded sections were immunolabeled with anti von Willebrand factor antibody (Dako, Glostrup, Denmark). Ten different fields were randomly selected, and the number of von Willebrand factor-positive cells in each field was counted using a light microscope under high-power magnification (×200).

The 3D-hiPSC-CT and whole implanted cardiac tissue were fixed with 4% paraformaldehyde and labeled with primary antibodies, including anti-cardiac troponin T (cTnT, 1:200 dilution; Abcam, Cambridge, UK), anti-sarcomeric alpha actinin (α-actinin, 1:400; Sigma), anti-connexin43 (1:100; Abcam), anti-fibronectin (1:200; Abcam), anti-collagen IV (1:100; Abcam), anti-heparan sulfate proteoglycan 2 (1:100; Abcam), anti-desmin (1:100; Abcam), and anti-dystrophin (1:50; Abcam), followed by incubation with the secondary antibodies AlexaFluor488- or AlexaFluor555-conjugated goat or donkey anti-mouse or anti-rabbit (ThermoFisher Scientific). Nuclei were counterstained with Hoechst33342 (Dojindo Molecular Technologies, Kumamoto, Japan). Staining patterns were observed using a confocal microscope (FLUOVIEW FV10i; Olympus, Tokyo, Japan).

## Ultrastructural analysis

The 3D-hiPSC-CT and resected rat heart were immediately fixed in 2.5% glutaraldehyde in 0.1 M phosphate buffer at 4°C and 1% osmium tetroxide, dehydrated with alcohol, and embedded in epoxy resin. We prepared 1 μm thick sections and stained them with toluidine blue for optical microscopy. We also prepared 70 nm thin sections and stained them with uranyl acetate and lead citrate. These sections were generated by the Center for Medical Research and Education, Graduate School of Medicine, Osaka University. Each section was examined and photographed using a Hitachi H-7650 transmission electron microscope (TEM) (Hitachi, Tokyo, Japan).

## Real-time polymerase chain reaction (PCR)

Total RNA was isolated from the peri-infarct zone of the cardiac tissue after hiPSC-CT implantation using RNeasy Fibrous Tissue Mini Kit (Qiagen, Hilden, Germany). Real-time PCR was then performed using the ViiA 7 RealTime PCR System (Thermo Fisher Scientific) with a TaqMan (Thermo Fisher Scientific) probe and rat-specific primers (Applied Biosystems) for VEGF (Assay ID: Rn01511601_m1) and HGF (Assay ID: Rn00566673_m1). All data were normalized using *GAPDH* as a control and evaluated using the delta-delta cycle threshold (Ct) method.

## Statistical analyses

JMP software (JMP pro13; SAS Institute Inc.) was used for all statistical analyses. Data are expressed as mean ± standard deviation. Statistical significance was determined by Student's *t*-test (two-tailed) for comparisons between two groups; $P < 0.05$ was considered statistically significant.

## Results

### FN-G-coated hiPSC-CMs had improved tolerance to hypoxia

After incubation of hiPSC-CMs in a hypoxia condition for 72 h, more LDH was released in the cells coated with FN-G, implying that cell injury had significantly decreased (P < 0.01; Fig 1B). In addition, the CCK8 assay showed that the FN-G coating significantly improved cell viability (P < 0.001; Fig 1C). Moreover, based on analysis with the Cell Motion Imaging System, the beating area of FN-G-coated hiPSC-CMs was found to be significantly broader than that of the uncoated control cells (P < 0.001; Fig 1D).

### The 3D-hiPSC-CT improved cardiac function in the myocardial infarction rat model

Preoperative heart function did not differ between the 3D-hiPSC-CT-implanted and sham control groups. Four weeks after transplantation, the ejection fraction of the 3D-hiPSC-CT-implanted group was significantly better than that of the control group (P < 0.001; Fig 3A).

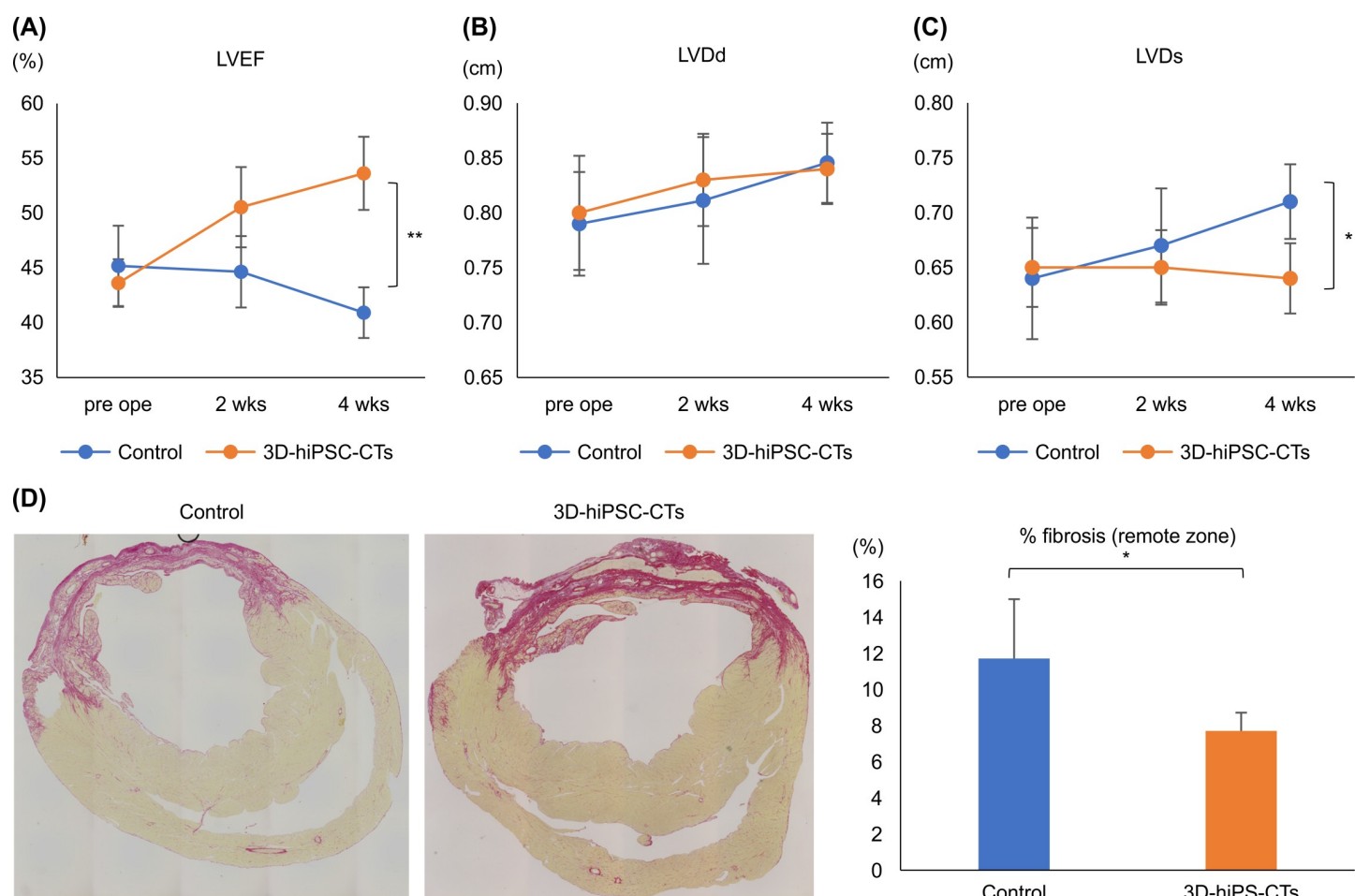

**Fig 3. The three-dimensional human induced pluripotent cell-derived cardiac tissue (3D-hiPSC-CT) improved cardiac function in the myocardial infarction rat model.** (A–C) Results of B-mode echocardiogram: left ventricular ejection fraction (LVEF), end-diastolic diameter (LVDd), and end-systolic diameter (LVDs); *P < 0.01, ** P < 0.001. (D) Fibrosis area at the remote zone in myocardial infarction hearts 4 weeks after transplantation. Sections were assessed by Picro Sirius Red staining. * P < 0.01.

Furthermore, the end-diastolic left ventricle diameter was not different between the two groups (P = 0.65; Fig 3B), whereas the end-systolic left ventricle diameter was significantly smaller in the 3D-hiPSC-CT group than that of the control group (P < 0.01; Fig 3C).

Four weeks after implantation, histological analysis showed that the fibrotic area of the whole cardiac tissue did not differ between the 3D-hiPSC-CT and control groups. However, the ratio of fibrosis in the remote zone was significantly smaller in the 3D-hiPSC-CT group than that in the control group (P < 0.01; Fig 3D).

## The 3D-hiPSC-CT induced angiogenesis and angiogenic cytokine expression at the peri-infarct zone

Hematoxylin and eosin staining revealed that the 3D-hiPSC-CT was maintained and had survived on the epicardium 4 weeks after transplantation (Fig 2D and 2E). The 3D-hiPSC-CT group showed a significantly enhanced capillary density in the peri-infarct zone compared with that of the control group (Fig 4A).

Positive expression of isolectin B4 (a marker of the dermis and vascular endothelial cells) was found in the implanted cardiac tissue, which indicated good angiogenesis to the graft (Fig 4B). In addition, real-time PCR showed that the relative mRNA expression levels of the angiogenic factors *VEGF* and human growth factor (*HGF*) in the peri-infarct zone were significantly higher in the 3D-hiPSC-CT group than those of the control group (Fig 4C).

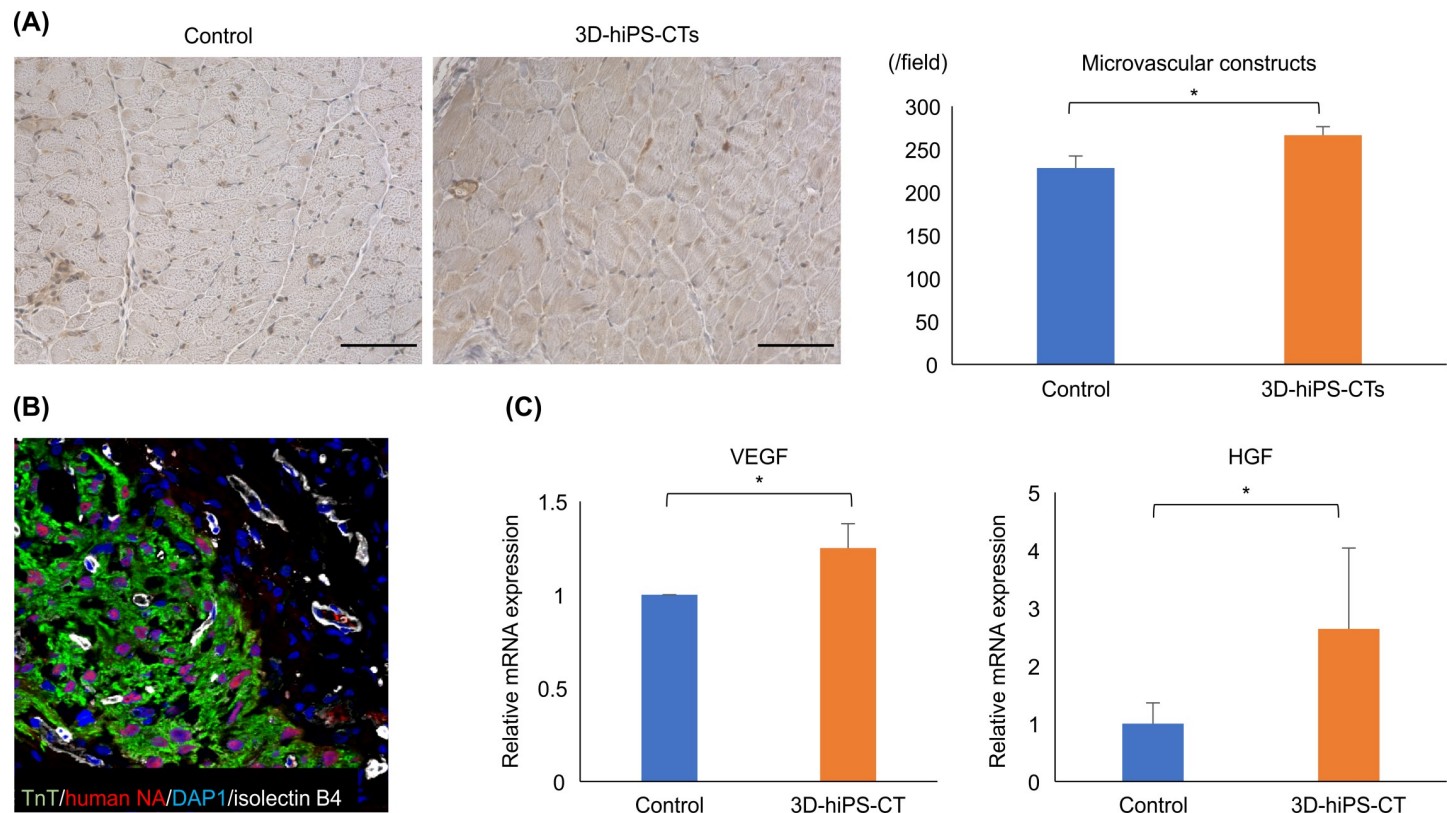

**Fig 4. The three-dimensional human induced pluripotent stem cell-derived cardiac tissue (3D-hiPSC-CT) induced angiogenesis and angiogenic cytokine expression in the peri-infarct zone.** (A) Capillary density in the peri-infarct zone in the myocardial infarction rat heart 4 weeks after transplantation; scale bar = 100 μm. Sections were assessed by immunohistochemical staining for von Willebrand factor; * P < 0.01. (B) Immunostaining for isolectin B4 (white), TnT (green), human nuclei (red), and DAPI (blue); scale bar = 50 μm. (C) Quantitative polymerase chain reaction analysis of angiogenic cytokine-related gene expression (*Vegf* and *Hgf*; * P < 0.01.

**(A)**

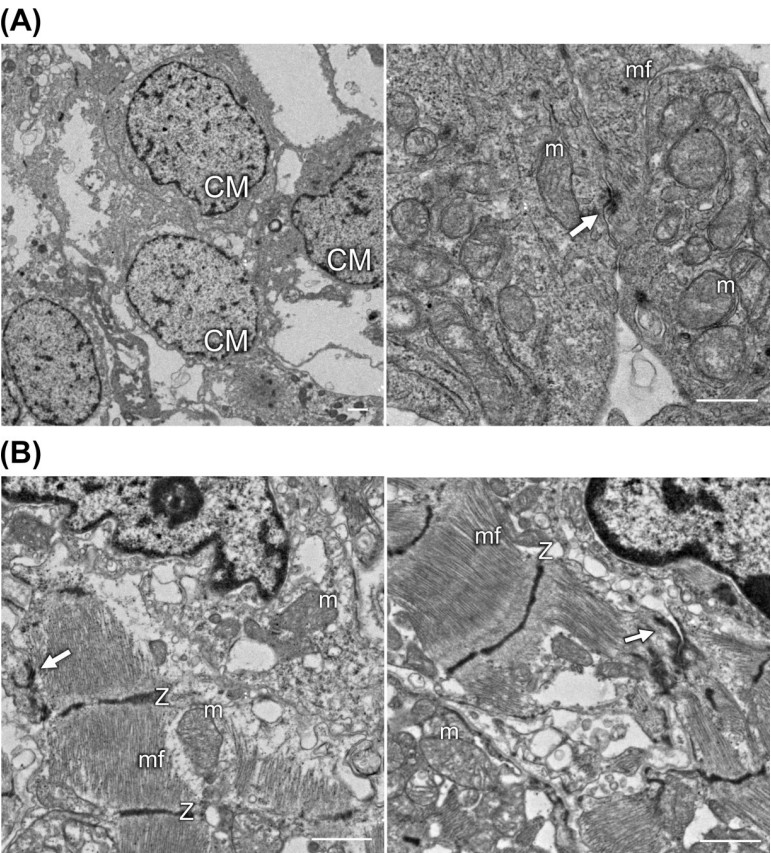

**(B)**

**Fig 5. Ultrastructural analysis of 3D-hiPSC-CT using transmission electron microscopy (TEM).** (A) TEM images of 3D-hiPSC-CT *in vitro*. The hiPSC-CM (CM) are assembled to be in contact with each other via desmosomes (white arrow). A few myofibrils (mf) and mitochondria (m) are observed within the cardiomyocytes, scale bar = 1 μm. (B) TEM images of 3D-hiPSC-CT *in vivo*. The 3D-hiPS-CT showed clear thickened myofibrils (mf), Z-band (Z), and mitochondria (m). The adherens junction (black arrow) and desmosome (white arrow) were also disclosed between the iPSC-CM, scale bar = 1 μm.

## Ultrastructural analysis of 3D-hiPSC-CT using transmission electron microscopy

Ultrastructural analysis of 3D-hiPSC-CT using transmission electron microscopy. In the 3D-hiPSC-CT, irregularly shaped hiPSC-CMs aggregate in several layers. Each manufactured cardiomyocyte contained some myofibrils and mitochondria with a few cristae. The cells contacted each other via desmosomes (Fig 5A).

The implanted 3D-hiPS-CT showed clear thickened myofibrils, Z-band, and mitochondria with many lamellar cristae. The adherens junction and desmosome were also disclosed between the iPSC-CM (Fig 5B).

## Implantation of 3D-hiPSC-CT promoted ECM remodeling and cardiomyocyte maturation

Although the expression of FN, which was used to construct the 3D-hiPSC-CT, was detected in the 3D cardiomyocytes *in vitro* (Fig 6A-1), it was only poorly expressed in the cardiac tissue of the rats 12 weeks after transplantation (Fig 6A-2). However, collagen type IV and perlecan localized in the basement membrane were clearly expressed in the transplanted tissue

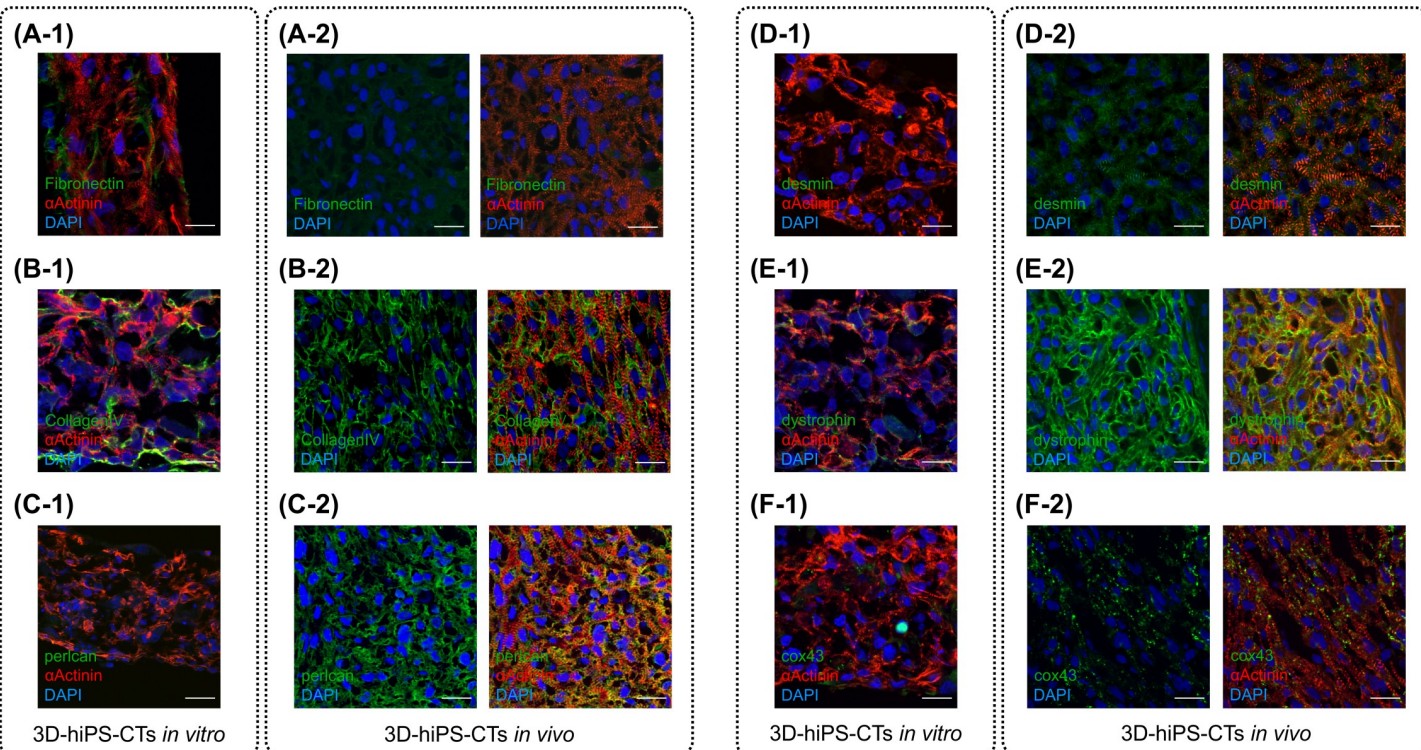

**Fig 6. Implantation of three-dimensional human induced pluripotent stem cell-derived cardiac tissue (3D-hiPSC-CT) promoted extracellular matrix (ECM) remodeling and maturation of cardiomyocytes.** (A-1) Immunostaining of fibronectin (green) on 3D-hiPSC-CT *in vitro*. (A-2) Fibronectin expression (green) was poor in the cardiac tissue 12 weeks after transplantation. (B-1) Immunostaining of collagen IV (green) on the 3D-hiPSC-CT *in vitro*. (B-2) Collagen type IV (green) was clearly expressed in the transplanted tissue 12 weeks after implantation. (C-1) Immunostaining of perlcan (green) was not observed on the 3D-hiPSC-CT *in vitro*. (C-2) Perlcan (green) was clearly expressed in the transplanted tissue 12 weeks after implantation. (D-1) Immunostaining of desmin (green) was not observed on the 3D-hiPSC-CT *in vitro*. (D-2) Desmin (green) was expressed in the transplanted tissue. (E-1) Immunostaining of dystrophin (green) was not observed on the 3D-hiPSC-CT *in vitro*. (E-2) Dystrophin (green) was clearly expressed in the transplanted tissue. (F-1) The expression of connexin43 (green) was poor on the 3D-hiPSC-CT *in vitro*. (F-2) Connexin43 (green) was clearly expressed in the transplanted tissue 12 weeks after implantation.

(Fig 6B-2 and 6C-2). The intermediate filaments desmin (Fig 6D-2) and dystrophin (Fig 6E-2), which stabilize the plasma membrane of striated muscle cells, were also found to be well expressed in the transplanted tissue; however, expression of these proteins was not detected in the tissue before implantation (Fig 6D-1 and 6E-1). Connexin43 was also clearly expressed in the implanted 3D-hiPSC-CT (Fig 6F-2).

## Discussion

The results of this study demonstrate that FN-G-coated cardiomyocytes may be substantially more resistant to hypoxia than uncoated cardiomyocytes. Moreover, the transplanted FN-G-coated 3D-hiPSC-CT was histologically detected up to 12 weeks after transplantation, which may contribute to cardiac function recovery. In particular, the expression of fibronectin in the original FN-G-coated 3D-hiPSC-CT was reduced, while the expression of other types of ECM that are more appropriate for improving cardiac performance was promoted in the transplanted 3D-hiPSC-CT after 12 weeks. The expression of cardiac proteins such as dystrophin, perlcan, or desmin was prominent in the cardiomyocytes, accompanied by connexin43 expression 12 weeks after implantation.

Angiogenesis in the early phase after transplantation is essential for the prolonged survival of transplanted myocardial tissue [4], in addition to appropriate ECM to promote the

construction of the myocardial tissue. In the transplanted tissue, cardiomyocytes bind to each other via the fibronectin coating to maintain tissue morphology along with induction of angiogenesis in the early phase after transplantation. However, after transplantation, other ECMs on basement membranes may be more essential to maintain the activity and engraft the transplanted cardiomyocytes.

The primary constituents of the basement membrane are collagen IV, laminin, nidogen, and perlecan. In particular, collagen IV and perlecan are essential for the mechanical stability of cardiomyocytes and their electrical conduction by binding to the surrounding interstitial ECM and stromal cells [14, 15]. In addition, laminin plays an essential role in controlling cell adhesion activity and cell death through cell signaling [16–18]. Thus, the ECM, which creates an appropriate microenvironment for myocardial tissue, is crucial for the successful engraftment of transplanted cardiomyocytes. In this study, although the FN-G-coated 3D-hiPSC-CT showed decreased fibronectin expression 12 weeks after transplantation, collagen IV and perlecan, which were not expressed on the basement membrane before transplantation, were prominently expressed following *in vivo* transplantation. The possible mechanism explaining this result may involve fibroblasts in the transplanted myocardial tissue that produce ECMs. Overall, these findings demonstrate that fibronectin provides a niche environment for hiPSC-derived cardiomyocytes to survive after transplantation and supports early angiogenesis to survive in a hypoxic condition. Furthermore, it is speculated that the basement membrane required for survival in the recipient heart tissue was generated from fibroblasts in the 3D-hiPSC-CT.

Although connexin43 was not expressed in the 3D-hiPSC-CT *in vitro* before transplantation, its expression was detected between the transplanted cardiomyocytes 12 weeks after transplantation. Fibronectin was reported to increase the level of connexin43 expression in alveolar epithelial cells [19], and laminin could increase the expression level of connexin in hippocampal progenitor cells and tracheal epithelial cells [20], suggesting that ECM may enhance electrical coupling between cells. We also found that dystrophin was poorly expressed in the *in vitro* 3D-hiPSC-CT before transplantation, but was detected along the cell membrane *in vivo* 12 weeks after transplantation. The dystrophin-glycoprotein complex, which binds to the ECM via myocardial cell membrane proteins, could also be activated though some signals from the ECM [21]. Thus, transplanting the 3D-hiPSC-CT into the heart could allow signals from the ECM to reach the cardiomyocytes via laminin or integrin receptor, thereby inducing the expression of certain proteins that are essential for maintaining the structure and physiology of cardiomyocytes.

In clinical settings, ECM coated 3D-hiPSC-CT has the potential to improve survival and therapeutic efficacy in ischemic heart disease. This strategy is a promising treatment option as an alternative to heart transplantation and left ventricular assist devices. It should be noted that this study included some limitations. Although we found good survival of 3D-hiPSC-CT and ECM remodeling surrounding the implanted tissue, the mechanism of ECM remodeling has not been elucidated. Notwithstanding, our study provides novel insights into cell viability of the ECM of the heart.

In conclusion, FN-G-coated 3D-hiPSC-CM tissues improved cardiac function in an ischemic heart failure rat model by remodeling cardiac proteins and the basement membrane matrix.

## Supporting information

**S1 Movie. Beating 3D-hiPSC-CTs during transplantation into rats.**
(MP4)

## Acknowledgments

The authors thank Akima Harada for excellent technical assistance.

## Author Contributions

**Conceptualization:** Takami Akagi, Mitsuru Akashi.

**Supervision:** Yoshiki Sawa.

**Writing – original draft:** Junya Yokoyama.

**Writing – review & editing:** Shigeru Miyagawa.

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
