## [Decision Letter · Decision Letter 0]

1 Feb 2021

PONE-D-20-41017

Human induced pluripotent stem cell-derived three-dimensional cardiomyocyte tissues ameliorate the rat ischemic myocardium by remodeling the extracellular matrix and cardiac protein phenotype

PLOS ONE

Dear Dr. Sawa,

Thank you for submitting your manuscript to PLOS ONE. After careful consideration, we feel that it has merit but does not fully meet PLOS ONE’s publication criteria as it currently stands. Therefore, we invite you to submit a revised version of the manuscript that addresses the points raised during the review process.

All issues raised by expert reviewers are required.

We look forward to receiving your revised manuscript.

Kind regards,

Vincenzo Lionetti, M.D., PhD

Academic Editor

PLOS ONE

Journal Requirements:

Reviewers' comments:

Reviewer's Responses to Questions

**Comments to the Author**

1. Is the manuscript technically sound, and do the data support the conclusions?

Reviewer #1: No

Reviewer #2: Yes

2. Has the statistical analysis been performed appropriately and rigorously? 

Reviewer #1: Yes

Reviewer #2: Yes

3. Have the authors made all data underlying the findings in their manuscript fully available?

Reviewer #1: Yes

Reviewer #2: Yes

4. Is the manuscript presented in an intelligible fashion and written in standard English?

Reviewer #1: Yes

Reviewer #2: Yes

5. Review Comments to the Author

Reviewer #1: This study investigates the beneficial effects of implantation of 3D hiPSC-cardiomyocytes sheets coated with FN and gelatin on the cardiac performance in MI rats. The authors assessed the cardiac function, graft survival and vascularization in control group and iPSC-CM group. Though the study design and the idea behind the 3D FN G coated iPSC-CT is interesting, however, the experimental design needs significant improvement before this study can be considered for publication. Following are my more specific comments.

1)What was the control used in the hypoxia and in vivo study? Page 9, line 134 the authors have mentioned uncoated, is this the right control to be used to compare the layer-by-layer generated iPSC-CT? Generally, for generation of CT one would use collagen, fibronectin or Matrigel?

2)Generally, in case of the cardiomyocytes maintained in media containing FBS, the FBS tend to prevent the beating of cells. Authors need to clarify this, how they were able to get beating cardiomyocytes for the experiment during transplantation?

3)What is the conformation test for the coating of fibronectin and gelatin on iPSC-CMs? Authors must perform analytical evaluations similar to SEM and FTIR and provide data to improve rigor of the current study.

4)There is no sham group in the experiment setup. Authors should include sham group during the analysis of the in vivo data.

5)Page 6 line 86 replace iPSC-CM as iPSC.

Reviewer #2: This manuscript has valuable information to scientists who involve in research of heart failure.Especially, as compared to other multipotent stem cells, such as ES cell, iPS cell has potential benefit by minimizing the risk of rejection, ethical problem, and tumorigenesis. It seems that creating iPS cell still has comlicated process that has to be improved, I think prompt clinical trial should be recommended.

6. PLOS authors have the option to publish the peer review history of their article (what does this mean?). If published, this will include your full peer review and any attached files.

Reviewer #1: No

Reviewer #2: No

---

## [Author Response · Author response to Decision Letter 0]

19 Feb 2021

Reviewer 1:

Q1.

What was the control used in the hypoxia and in vivo study? Page 9, line 134 the authors have mentioned uncoated, is this the right control to be used to compare the layer-by-layer generated iPSC-CT? Generally, for generation of CT one would use collagen, fibronectin or Matrigel?

A1.

Response: Thank you for your comment. The FN/G coated cells and uncoated cells were compared in the hypoxia experimental system. 

In vitro, when 3 × 106 cells were placed in a 24-well cell culture insert without coating, tissues could not be created as the cells became heterogeneous. Thus, we were not able to compare coated and uncoated 3D-hiPSC-CTs.

Additionally, when 3D-hiPSC-CTs were generated, we used only fibronectin and gelatin—not collagen—as a small population of vimentin-positive cells are found in iPS induction; there is a possibility that collagen derived from fibroblasts may be present.

Q2.

Generally, in case of the cardiomyocytes maintained in media containing FBS, the FBS tend to prevent the beating of cells. Authors need to clarify this, how they were able to get beating cardiomyocytes for the experiment during transplantation?

A2.

Response: Thank you for pointing this out. In this study, 3D-iPS-CTs were cultured in a medium containing 10% FBS serum. We confirmed that the cardiomyocytes were beating in this experimental system daily and that the myocardial tissue itself was beating at the time of transplantation. FBS was flushed out avoid immunological rejection prior to CT transplantation. We have included the relevant sentences in the manuscript.

Q3.

What is the conformation test for the coating of fibronectin and gelatin on iPSC-CMs? Authors must perform analytical evaluations similar to SEM and FTIR and provide data to improve rigor of the current study.

A3.

Response: We appreciate the helpful suggestion. As suggested, we have presented the TEM image.

We have included the following: “Ultrastructural analysis of 3D-hiPSC-CT using transmission electron microscopy. In the 3D-hiPSC-CT, irregularly shaped hiPSC-CMs aggregate in several layers. Each manufactured cardiomyocyte contained some myofibrils and mitochondria with a few cristae. The cells contacted each other via desmosomes (Fig 5A).

The implanted 3D-hiPS-CT showed clear thickened myofibrils, Z-band, and mitochondria with many lamellar cristae. The adherens junction and desmosome were also disclosed between the iPSC-CM (Fig 5B).”

In regards to the conformation test for coating, the iPS-CM was coated by the method by our co-author Akashi [1], which confirmed coating; thus, we had not confirmed the results of coating again.

[1] Nishiguchi A, Matsusaki M, Miyagawa S, Sawa Y, Akashi M. Dynamic nano-interfaces enable harvesting of functional 3D-engineered tissues. Adv Healthc Mater. 2015;4(8): 1164-1168. doi:10.1002/adhm.201500065, PubMed: 25728509.

Q4.

There is no sham group in the experiment setup. Authors should include sham group during the analysis of the in vivo data.

A4.

Response: In vitro, when 3 × 106 cells were placed in a 24-well cell culture insert without coating, tissues could not be created as the cells became heterogeneous. Thus, the transplant experiment was a comparative experiment between sham control—which had no transplanted tissue—and the 3D-hiPS CT transplant.

Q5. 

Page 6 line 86 replace iPSC-CM as iPSC.

A5.

Response: Thank you, as suggested we have revised iPSC-CM to iPSC.

Reviewer2:

Thank you for your peer review despite your busyness. We will continue to strive for the development of the field of regenerative medicine.

---

## [Decision Letter · Decision Letter 1]

23 Feb 2021

Human induced pluripotent stem cell-derived three-dimensional cardiomyocyte tissues ameliorate the rat ischemic myocardium by remodeling the extracellular matrix and cardiac protein phenotype

PONE-D-20-41017R1

Dear Dr. Sawa,

We’re pleased to inform you that your manuscript has been judged scientifically suitable for publication and will be formally accepted for publication once it meets all outstanding technical requirements.

Kind regards,

Vincenzo Lionetti, M.D., PhD

Academic Editor

PLOS ONE

Additional Editor Comments (optional):

Reviewers' comments:

Reviewer's Responses to Questions

**Comments to the Author**

1. If the authors have adequately addressed your comments raised in a previous round of review and you feel that this manuscript is now acceptable for publication, you may indicate that here to bypass the “Comments to the Author” section, enter your conflict of interest statement in the “Confidential to Editor” section, and submit your "Accept" recommendation.

Reviewer #1: All comments have been addressed

Reviewer #2: All comments have been addressed

2. Is the manuscript technically sound, and do the data support the conclusions?

Reviewer #1: Yes

Reviewer #2: Yes

3. Has the statistical analysis been performed appropriately and rigorously? 

Reviewer #1: Yes

Reviewer #2: Yes

4. Have the authors made all data underlying the findings in their manuscript fully available?

Reviewer #1: No

Reviewer #2: Yes

5. Is the manuscript presented in an intelligible fashion and written in standard English?

Reviewer #1: Yes

Reviewer #2: Yes

6. Review Comments to the Author

Reviewer #1: Authors have addressed all my concerns. The manuscript has improved significantly. I have no further comments comments.

Reviewer #2: I think asuthors have addressed all the required change suggested in previous review. I would like to encourage iPS study to be available in real clinical setting soon.

7. PLOS authors have the option to publish the peer review history of their article (what does this mean?). If published, this will include your full peer review and any attached files.

Reviewer #1: No

Reviewer #2: No

---

## [Editor Report · Acceptance letter]

5 Mar 2021

PONE-D-20-41017R1 

Human induced pluripotent stem cell-derived three-dimensional cardiomyocyte tissues ameliorate the rat ischemic myocardium by remodeling the extracellular matrix and cardiac protein phenotype 

Dear Dr. Sawa:

I'm pleased to inform you that your manuscript has been deemed suitable for publication in PLOS ONE. Congratulations! Your manuscript is now with our production department. 

Kind regards, 

on behalf of

Prof. Vincenzo Lionetti 

Academic Editor

PLOS ONE